# One-Temperature Analytical Model for Femto-/Atto-Second Laser–Metals Drilling: A Novel Approach

**DOI:** 10.3390/ma15145010

**Published:** 2022-07-19

**Authors:** Cristian N. Mihailescu, Muhammad Arif Mahmood, Natalia Mihailescu, Mihai Oane

**Affiliations:** 1Laser Department, National Institute for Laser, Plasma and Radiation Physics (INFLPR), 077125 Magurele, Romania; cristi.mihailescu@inflpr.ro; 2Mechanical Engineering Program, Texas A&M University at Qatar, Doha P.O. Box 23874, Qatar; muhammad.arif_mahmood@outlook.com; 3Accelerators Laboratory, National Institute for Laser, Plasma and Radiation Physics (INFLPR), 077125 Magurele, Romania; mihai.oane@inflpr.ro

**Keywords:** ultrashort laser pulses, non-Fourier heat equation, thermal waves, metals, modelling

## Abstract

Recently, ultrafast lasers have been developed and potentially become a point of interest worldwide, as their interaction with matter is yet unknown and can be mediated by new physical mechanisms. Real-time experimentation requires enormous costs, and there is therefore a need to develop computational models for this domain. By keeping in view this idea, a non-Fourier heat equation has solved the case of ultrafast laser–material interaction. Initial and boundary conditions were considered, and a one-dimensional mathematical model was presented. The simulations were compared with the experimental results for ultrashort laser–metallic sample interaction, and a close correlation was proven. It was found that the coupling of electron–phonon becomes “zero” due to short laser–material interaction time. The propagation of thermal waves was identified due to non-Fourier heat implementation. When the pulse duration increases, the variation in the thermal distribution becomes trivial due to an inverse correlation between the pulse duration and total energy within the pulse. When the laser–material interaction time decreases from fs to as, the generation of thermal waves increases and the powerful laser intensity acts as a shock wave during laser–material interaction, which causes a higher intensity of the thermal wave.

## 1. Introduction

Recently, the ultrashort-pulsed laser–metal interaction has gained high interest, owing to rapid energy deposition with smaller heat-affected zones during metal processing with femtosecond lasers [1,2,3]. To explain ultrashort pulse laser ablation, different studies utilizing phase and coulomb explosions [4,5] and thermoelastic-wave effects [6], have been proposed. The primary mechanism, according to Du et al. [7], consists of three stages: (a) laser energy absorption by photon–electron coupling, (b) energy distribution to the lattice via electron–phonon interaction, and (c) normal energy diffusion into the material via phonon–phonon collision. According to the two-temperature model (TTM), the electron–phonon coupling has long been thought to have a role in processing metallic objects when ultrashort laser pulses are used for ablation [2,8]. The electron–lattice relaxation timeframe is typically several picoseconds, while the laser pulse is just a few hundred fs long. When exposed to sub-ps laser pulses, the metallic material is stimulated into a high non-equilibrium state [9].

A TTM was used to explain the ultrashort laser-metal processing mechanism. This continuous model depicts energy transmission inside a metal lattice with two coupled generalized heat conduction equations in the case of electrons and phonons temperatures. Using a single fs laser pulse to heat thin metal sheets has been theoretically and practically documented. Thus, Li et al. [10] devised a hybrid model that combined the standard TTM with molecular dynamics to view the real ablation process at an atomic scale. Li et al. [10] simulated three-dimensional (3D) ablation craters produced by a single pulse with variable laser energy. However, the fundamental energy transmission mechanism for multi-pulse fs laser heating is still unknown.

Furthermore, the TTM has been widely employed by researchers and scientists to understand better the electron–phonon energy transfer in the case of thin metallic sheets [11,12,13,14]. In most of the investigations, the Gaussian laser beam spot size has been assumed as constant. However, this assumption may lead to erroneous findings when using multi-pulse lasers for ablation, particularly when drilling holes at a micro-scale with a high aspect ratio (10:1), where the spot size is a function of depth. Another model is required to investigate energy transport inside a thick layer by multi-pulse fs laser ablation.

In a study by Buca et al. [15], a classical Anisimov’s TTM was linked with the 3D telegraph Zhukovsky equation. This technique was used to calculate the electron temperature field during the fs laser pulse–gold interaction. The interaction between the laser pulse and the metal sample during the initial ps was found to be governed by the relaxation time and electron–phonon coupling factor. Oane et al. [16] applied the Zhukovsky formalism to a specific experimental setting, namely a one-dimensional (1D) lattice made up of Au nanoparticles with a radius of 20 nm in aqueous media irradiated with a 20 ns laser pulse. This method simulated the thermal fields of carbon nanoparticles implanted in a vitreous matrix triggered by laser pulses and electrical fields.

A non-Fourier heat equation was solved in the case of ultrafast laser-material interaction by transforming TTM to a one-temperature model (OTM) and is reported in this study. Theoretically, at atto-/femto-seconds laser pulses–material interaction, there is no interaction between electrons and phonons [17]. Hence, it does not make sense to imply TTM. Accordingly, OTM has been deduced and brings the novelty in the as/fs laser pulses–material interaction. Only a few studies [18,19] were carried out for as/fs laser pulses–material interaction. These studies were, however, based on numerical and semi-numerical approaches. Such techniques require high computational time due to the iterative computational technique, and the solution accuracy is dependent on the mesh quality [20]. A finer mesh requires long computational time [21]. To avoid such difficulties, an analytical model has been presented herewith. Initial and boundary conditions were applied to calculate the final expression for ultrashort-pulse to material interaction. A comparison was made between experiments and simulations to identify the trustworthiness of the developed solution.

## 2. Mathematical Formalism

The particularized form of the TTM is expressed as:(1)$$\left(\partial_{t}^{2}+\varepsilon \partial_{t}\right)T\left(x,y,z,t\right)=\left(\alpha \partial_{x}^{2}+\beta \partial_{y}^{2}+\gamma \partial_{z}^{2}+k\right)T\left(x,y,z,t\right).$$

It has been considered that the coupling factor between electrons and phonons is zero due to the short laser–material interaction time. In Equation (1), ε is a constant; *x*, *y*, *z* and t are the Cartesian coordinates in space and time; α=β=γ are thermal conductivity of electrons; *k* is the coupling factor between electron and phonon, equal to zero; and T is the temperature. It is worth noting that the heat source term has been included in Equation (1) via initial boundary conditions, as given in Equation (2):(2)$$T\left(x,y,z,0\right)=\frac{1-R}{\delta +\delta_{b}}F\frac{w_{o}^{2}}{w^{2}\left(z\right)}exp\left(-\frac{z-z_{s}}{\delta +\delta_{b}}-\frac{2{\left(x-x_{o}\right)}^{2}+2{\left(y-y_{o}\right)}^{2}}{w^{2}\left(z\right)}\right).$$

Here, *R* is the metal’s reflectivity, δ is the optical penetration of the laser beam within the target, $\delta_{b}$ is the ballistic length, *F* is the laser fluence, *w_o_* is the waist of the laser beam, and *w*(*z*) is the waist of the laser beam as a function of the propagation axis. The final boundary for time is expressed as:(3)T(x,y,z,∞)=0.

Here, note that *T* is a temperature variation and not the absolute value. After replacing the ε; α, β, γ; and *k* with $C_{e}$; $K_{e}$; and “*g*”, respectively, one can transform Equation (1) into the following:(4)$$\frac{\partial^{2}T_{e}}{\partial t^{2}}+C_{e}\frac{\partial T_{e}}{\partial t}=K_{e}\left(\frac{\partial^{2}T_{e}}{\partial x^{2}}+\frac{\partial^{2}T_{e}}{\partial y^{2}}+\frac{\partial^{2}T_{e}}{\partial z^{2}}\right)-gT_{e};\;T_{e}\gg T_{l}.$$

Here, $C_{e}$ is the heat capacity of electrons, $K_{e}$ the thermal conductivity of electrons, and “*g*” is the coupling factor between electron and phonon. One can observe that electron temperature (*T_e_*) is much higher than lattice temperature ($T_{l}$). When the laser beam pulse duration is concise, the electrons do not have enough time to elevate the phonons’ temperature, leading “*g*” to equal zero. After considering one pulse ($z_{s}=0$), Equation (2) can be re-defined by Equations (5)–(7):(5)$$T\left(x,y,z,0\right)=\frac{\left(1-R\right)F}{\delta +\delta_{b}}exp\left(-\frac{z}{\delta +\delta_{b}}-\left(\frac{2x^{2}+2y^{2}}{w_{o}^{2}}\right)\right).$$
(6)z→0 and y→0.
(7)$$T\left(x,y,z,0\right)\to T\left(x,0,0,0\right)=\frac{F}{\delta +\delta_{b}}exp\left(-\frac{z}{\delta +\delta_{b}}-\left(\frac{2x^{2}}{w_{o}^{2}}\right)\right).$$

The source term in the Zhukovsky formalism is given as [22]:(8)$$\sum_{n,\gamma }e^{\gamma x}x^{n};\;\gamma =-1\;\mathrm{and}\;n=0.$$

Here, γ is the coefficient of *z* (Equation (7)). Using Equations (7) and (8), Equation (9) has been deduced:(9)$$T\left(\xi ,t\right)=T\left(x,t\right)=t\frac{e^{-\frac{t\varepsilon }{2}+\gamma }}{4\sqrt{\pi }}\int_{0}^{\infty }\frac{due^{-\frac{t^{2}}{16u}-u\delta }}{u^{3/2}}H_{n}\left(\xi -2\gamma \tau_{,}-\tau \right).$$
(10)$$\delta =\varepsilon^{2}+4\left(K+\alpha \gamma^{2}\right).$$

In Equation (9), $H_{n}$ is the Hermite function of two variables, and τ (=4uα) is an increment based on the thermal conductivity. In any case, we have *n* = 0 that will give ($H_{n}\left(\xi -2\gamma \tau_{,}-\tau \right)$) equal to 1. The final solution for electron temperature in the case of as/fs single-pulse–metal interaction is:(11)$$T\left(x,t\right)=t\frac{e^{-\frac{t}{2}C_{e}-\frac{2x^{2}}{w_{o}^{2}}}}{4\sqrt{\pi }}\int_{0}^{\infty }\frac{du}{u^{3/2}}e^{-\frac{t^{2}}{16u}-u\left(C_{e}{}^{2}+4\left(k_{e}\right)\right)}.$$

## 3. Material and Methods

In this study, laser–aluminum (Al) interaction simulations were carried out [23] (Table 1).

Furthermore, to validate the developed model, experimental results from Ref. [23] have been used. A user-defined script file was written in MATHEMATICA software using Core i7, 8th Generation, with 16 GB Ram computing system. The computational time was approximately 20 s.

## 4. Results and Discussion

To identify the trustworthiness of the developed model, the simulation results were compared with the experimental ones in Ref. [23]. A close correlation has been identified between the experimental and simulation results with a deviation of 3.1%, as shown in Figure 1. This error has been calculated using the mean absolute deviation formula for the current simulation and published experimental results.

Figure 2a–d shows the thermal distribution within the Al sample in the case of 1, 25, 50, and 75 fs, respectively. The laser beam was placed in the middle of the sample (*x*-axis) and allowed to irradiate for the given time. All figures show a thermal wave propagating, designated as a change in thermal distribution, from the location of laser–Al interaction. It is due to the inherent characteristics of the non-Fourier heat equation. The temperature for all simulations, although indicated in arbitrary units, is self-consistent, so one can easily compare the results. However, one can observe that as the pulse duration increases, the change in the thermal distribution becomes insignificant in the case of laser–Al interaction. The average power in a provided pulse can be expressed as:(12)$$P_{avg}=\frac{P_{peak}\tau }{\Delta t}.$$

Here, $P_{avg}$ is the average power, τ is the measure of the time between the beginning and end of the pulse typically based on the full width half maximum (FWHM) of the pulse shape, $P_{peak}$ is the peak power, and Δt is the time between the start of one pulse and the start of the next. It is worth mentioning that as Δt decreases, the total energy of pulse width in FWHM increases. However, one should note that the Al sample is utilized during simulation analyses, having a limited laser beam absorption coefficient. In the beginning, the $P_{avg}$ increases by decreasing the Δt, resulting in higher thermal distribution change; however, the increment becomes insignificant even when the Δt decreases further due to the limited laser beam absorptivity of the Al-sample. This is why the change in thermal distribution becomes almost flat as the Δt increases.

Figure 3a–c shows the thermal distribution within the Al sample in the case of 1, 10, and 100 as, respectively. As the laser–Al interaction time decreased from fs to as the number of thermal waves increases. While shifting from fs to as regime, the laser intensity is more powerful. It produces an explosion upon impact with the sample surface, as the optical energy cannot be converted into thermal or vibrational energy in the as time frame. The pulse, in return, generates a thermal wave with a higher magnitude compared to the fs regime. The explanation, provided for Figure 2 using Equation (12), applies also here: the higher the pulse duration, the lower the total energy of a pulse in FWHM. This behavior can be clearly observed while shifting from 1 to 10 as and 10 to 100 as. One can analyze the higher number of thermal waves in Figure 3a compared to Figure 3b,c. The summation of thermal waves in Figure 3a results in a higher peak magnitude than the rest of the presented graphics.

Figure 4a–c compiles the thermal distribution across the depth of the Al sample irradiated by 1, 25, and 50 fs laser pulses, respectively. It can be seen that the thermal distribution is maximum at the laser–Al interaction area and declines rapidly to zero at a very close location with respect to the sample’s top surface. This is due to the very low probability of transferring the heat from electrons to the lattice during one laser pulse because the “*g*” term is equal to zero.

## 5. Conclusions

In this study, ultrafast laser–material interaction has been studied from the thermal point of view. For this purpose, the non-Fourier heat equation has been solved by considering the initial and boundary conditions. Following on, formalism has been deduced from one-dimensional and is easy to understand and use. The simulation results were compared with the experimental ones, and a close correlation was identified. The following conclusions were deduced from the analyses:It is feasible to consider the coupling of the electron–phonon has been deemed “zero” due to the very short laser–material interaction time. The precision of this assumption can be determined by achieving a close accuracy between the simulation and experimental results, with a deviation of 3.1%;Non-Fourier heat equations can show thermal wave propagation in laser–material interaction. Furthermore, it was identified that as the pulse duration increases, the change in the thermal distribution becomes insignificant due to an inverse correlation between the pulse duration and the total energy of the pulse;Moreover, when the laser pulse decreases from fs to as, the generation of thermal waves in Al sample increases, thus generating shock waves into the irradiated sample. It can be attributed to the generated powerful intensity that acts as a shock wave during laser–material interaction, which causes a higher magnitude thermal wave;It has been found that the thermal distribution is higher at the laser–material interaction area and declines depending on the material’s laser beam absorption coefficient;This study gives a time- and cost-effective method to estimate thermal distribution within the metallic sample in the case of ultrafast lasers–material interaction before experiments. It will assist the experimentalist in optimizing process parameters before experimentation, reducing the costs;More than these, the results of this study will help in the laser processing of metals through the rigorous design of the experimental parameters. By changing the metal constants, this model can be used for any metal, such as aluminum, iron, gold, copper, or silver;Previously, experiments had been carried out for the fs and ps laser-drilling of metals in Refs. [24,25]. The current model goes a step further by formulating a novel mathematical model (using Hermite polynomials of two variables) [26] of heat equation toward fs/as laser–metals drilling. The developed model will assist experimentalists in designing their experiments based on the simulation analyses before carrying out real-time experiments, thus reducing the trial-and-error cost involved.

## Figures and Tables

**Figure 1 materials-15-05010-f001:**
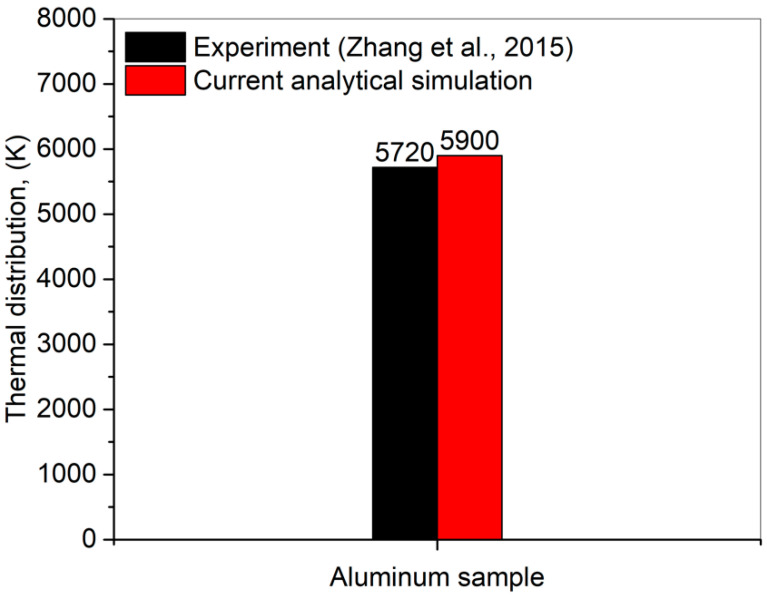
Thermal distribution for a single pulse comparison between the current analytical simulation model and experimental [23] results.

**Figure 2 materials-15-05010-f002:**
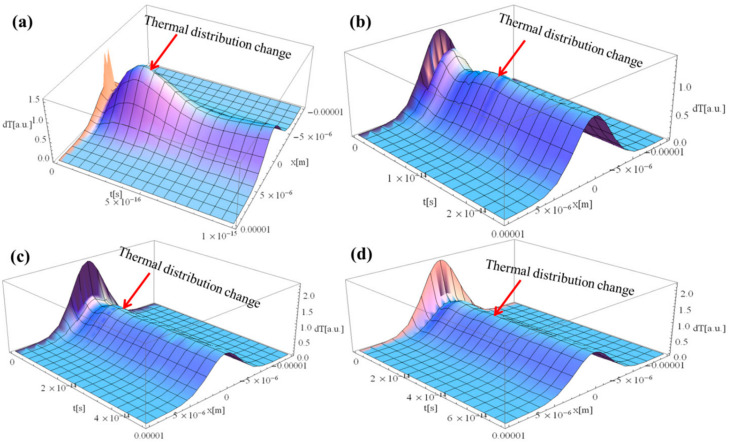
Laser–Al interaction for different duration of the laser pulse: (**a**) 1, (**b**) 25, (**c**) 50, and (**d**) 75 fs.

**Figure 3 materials-15-05010-f003:**
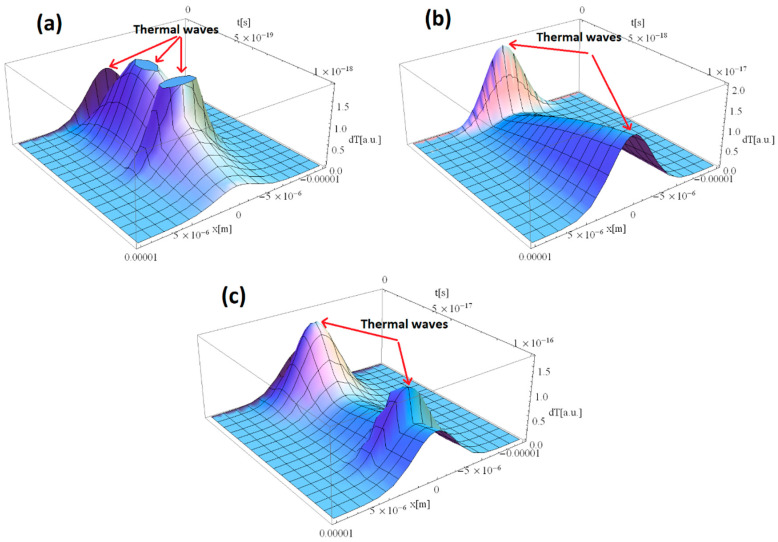
Laser–Al interaction for different durations of the laser pulse: (**a**) 1, (**b**) 10, and (**c**) 100 as.

**Figure 4 materials-15-05010-f004:**
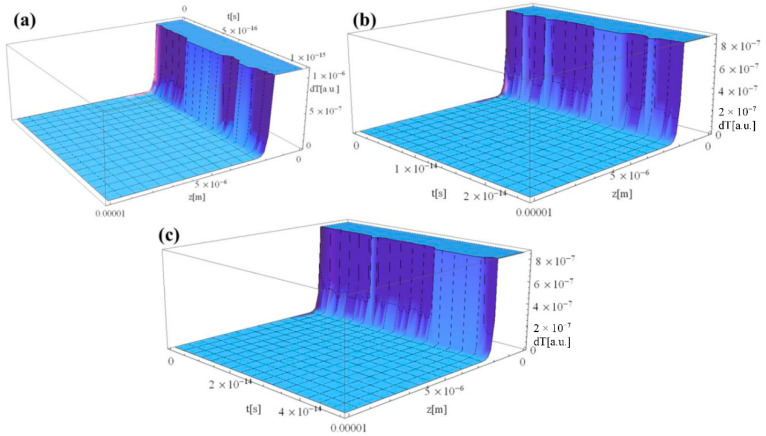
Laser–Al interaction along the sample depth (*z*-axis) for different durations of the laser pulse: (**a**) 1, (**b**) 25, and (**c**) 50 fs.

**Table 1 materials-15-05010-t001:** Parameters used as input in the simulation model.

Item	Numerical Values (Unit)
Thermal conductivity of electron	235 L/(mKs)
Volumetric heat capacity	134.5 J/(m^3^K)
Coupling factor between electrons and phonons	0
Rlectivity	0.88
Optical depth within the sample	20 nm
Ballistic length	100 nm
Room temperature	300 K

## Data Availability

Not applicable.

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
