# Peer review of "One-Temperature Analytical Model for Femto-/Atto-Second Laser–Metals Drilling: A Novel Approach"

_materials, 2022, doi:10.3390/ma15145010_

Round 1

Reviewer 1 Report

In Fig. 1, there is only one group of experimental and simulation results, which cannot fully explain the reliability of the model. Can the article add more experimental results? In addition, please add experimental setup descriptions.

Please explain why, in Figure 2, the change in heat distribution does not matter as the pulse time increases.

In Fig. 2 and Fig. 3, why does the heat distribution present this rule with the change of time?  Can it be explained?

Reviewer 2 Report

This submission cannot be accepted for publication for several reasons:

1. The presented "mathematical formalism" is very hard to understand, if the reader does not work on the same topic.

2. It is not made clear how the presented results (figures) correlate with the "mathematical formalism".

3. The presentation is not good; one citation is not contained in the references, these are not presented uniformly, there are many English mistakes and many typos.

Reviewer 3 Report

The article discusses about a novel analytical model for femto-/atto-second laser drilling, which could be interesting to the readers. The paper could be accepted after following minor revisions are made.

1. The manuscript has minor language-related typo/errors which should be corrected.

2. Recent papers (even from ~2022) in the field should be briefly reviewed and cited.

3. Impact of the work should be clarified. Ideas on how the simulation and theoretical results will directly help the laser-metals drilling process should be mentioned. Recommendations on experiments and laser-metals drilling should be made based on the simulation and theoretical results.

4. Applicability of the simulation and theoretical results to different/various (types of) metals and laser types should be elaborated.

Reviewer 4 Report

Real-time experimentation requires huge costs and there is a need to develop computational models for this domain. By keeping in view this idea, a non-Fourier heat equation has been solved in the case of ultrafast laser-material interaction. Initial and boundary conditions were involved, and a one-dimensional mathematical model was presented. The simulations were compared with the experimental results for ultrashort laser-Aluminum sample interaction and a close correlation was determined. The work is good, and it is a new point and I recommend accepting it, but there are some points that must be taken into consideration first to improve the paper.

1-There is an error in the affiliation numbering.

2-In line 18: the abbreviation should be defined where (laser-Al).

3- In line 86: check the capital and the small letters.

4-In lines 88-91: Please move the part describing Equation No. 4 after Equation No. 3.

5- Equation number 9 must be mentioned in line 108 (Eq. 9).

6-Is it possible to put the Figures (4-a, 4-b, and 4-c) in a new one Figure in order to facilitate the comparison process? And also, in Figures 2 and 3. It is possible to choose the most influential values in the figures (a, b, and c) and collect them in a new one figure for ease of comparison.

7-In the Conclusions, lines 175 to 177, how was the percentage of 4% calculated? Please check the value.

Round 2

Reviewer 1 Report

The revised manuscript has been greatly improved. I think the results and discussion as well as the English expression need to be further supplemented and improved, and then we can consider whether to accept it.

Reviewer 2 Report

The submission is almost ready for publication, however some points still have to be revised:

1. In the text, Sonntag et al. [10] and Li et al. [10] appear, and Sonntag et al. do not appear in the references.

2. The volumetric heat capacity has the unit J/(m^3K), and not J/(m^3K^2) as given in table 1.

3. Why are there "-g" and "g", and what for are the "-" for, anyway?

4. Why does dT in Figs. 2, 3, and 4 have arbitrary units? So how can the results be compared?

5. There are at least 21 interspaces (blanks) missing.
